# Uncovering the Topology of Time-Varying fMRI Data using Cubical Persistence

**Bastian Rieck**[*]
Dept. Biosystems (D-BSSE)
ETH Zurich & Swiss Institute
of Bioinformatics, Switzerland
`bastian.rieck@bsse.ethz.ch`

**Tristan Yates**[*]
Dept. of Psychology
Yale University
New Haven, CT, USA
`tristan.yates@yale.edu`

**Christian Bock**
Dept. Biosystems (D-BSSE)
ETH Zurich & Swiss Institute
of Bioinformatics, Switzerland
`christian.bock@bsse.ethz.ch`

**Karsten Borgwardt**
Dept. Biosystems (D-BSSE)
ETH Zurich & Swiss Institute
of Bioinformatics, Switzerland
`karsten.borgwardt@bsse.ethz.ch`

**Guy Wolf**
Dept. of Math. and Stat.
Univ. de Montréal; Mila
Montreal, QC, Canada
`guy.wolf@umontreal.ca`

**Nicholas Turk-Browne**[†]
Dept. of Psychology
Yale University
New Haven, CT, USA
`nicholas.turk-browne@yale.edu`

**Smita Krishnaswamy**[†]
Depts. of Gene. & Comp. Sci.
Yale University
New Haven, CT, USA
`smita.krishnaswamy@yale.edu`

## Abstract

Functional magnetic resonance imaging (fMRI) is a crucial technology for gaining insights into cognitive processes in humans. Data amassed from fMRI measurements result in volumetric data sets that vary over time. However, analysing such data presents a challenge due to the large degree of noise and person-to-person variation in how information is represented in the brain. To address this challenge, we present a novel topological approach that encodes each time point in an fMRI data set as a *persistence diagram* of topological features, i.e. high-dimensional voids present in the data. This representation naturally does not rely on voxel-by-voxel correspondence and is robust to noise. We show that these time-varying persistence diagrams can be clustered to find meaningful groupings between participants, and that they are also useful in studying within-subject brain state trajectories of subjects performing a particular task. Here, we apply both clustering and trajectory analysis techniques to a group of participants watching the movie 'Partly Cloudy'. We observe significant differences in both brain state trajectories and overall topological activity between adults and children watching the same movie.

## 1 Introduction

Human cognitive processes are commonly studied using functional magnetic resonance imaging (fMRI), amassing highly complex, well-structured, and time-varying data sets across multiple individual subjects. fMRI uses blood oxygen measurements of 3D brain volumes divided into *voxels*, i.e. 3D pixels with dimensions in the mm range. Voxels are measured over time while participants

---

[*]These authors contributed equally.
[†]These authors jointly supervised this work; corresponding authors.

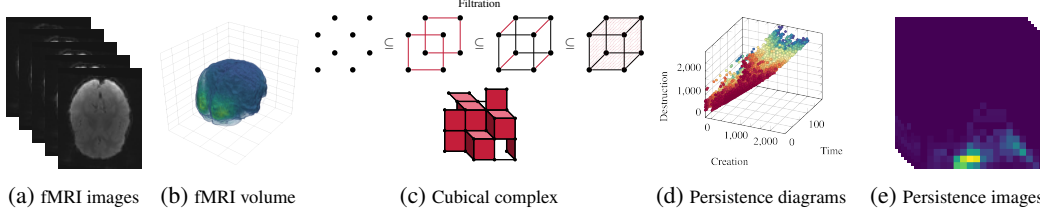

(a) fMRI images    (b) fMRI volume    (c) Cubical complex    (d) Persistence diagrams    (e) Persistence images

Figure 1: A graphical overview of our method. We represent an fMRI stack (a) as a volume (b), from which we create a sequence of cubical complexes (c). Calculating the persistent homology of this sequence results in a set of time-varying persistence diagrams (d); note that we only show the diagrams for a single dimension of the cubical complex. We calculate summary statistics from the diagrams (not shown), and convert them to vectorial representations (e) for analysis tasks.

perform cognitive tasks, resulting in time-varying activity measurements and an activation function over the volume. The ultimate goal of extracting higher-level abstractions from such data is primarily impeded by two factors: (i) the measurements are inherently noisy, due to changes in machine calibration, spurious patient movements, or environmental conditions, and (ii) there is a high degree of variability even between otherwise healthy brains (e.g. in terms of the representation of stimulus and activity in the brain). While these factors can be mitigated by certain experimental protocols and pre-processing decisions, they cannot be fully eliminated. This demonstrates the need for using representations that are to some extent *robust* with respect to noise and *invariant* with respect to isometric transformations in order to better capture cognitively-relevant fMRI activity, particularly across populations where anatomy–function relations may differ.

Traditional approaches largely ignore these factors, considering them inevitable noise in the measurements. Voxel activity is often either directly compared across different cognitive tasks, or the time-varying activity of voxels in pre-defined brain regions sharing functional properties is correlated to create a 'functional connectivity' graph. Our approach differs from existing approaches for fMRI data analysis in two crucial ways, namely (i) it is coordinate-free, providing a stable representation of high-level brain activity, even without a voxel-by-voxel match, and (ii) it does *not* require the creation of a correlation graph, or operate on any other approximated graph structure (in contrast to the MAPPER algorithm [55], for example). Instead, our method uses the 'raw' voxel activations themselves as a cubical complex, which we further characterise using time-varying persistence diagrams that indicate the presences of topological features such as voids of various dimensions in the voxel activations. These topological features are naturally invariant to a variety of shifts and noise (see Section 4 for more details). Our formulation enables the non-parametric analysis of fMRI data both statically and dynamically, i.e. for assessing differences between cohorts across time, and enabling insights into time-varying topological brain state trajectories within cohorts or individuals. For individuals, we calculate an averaged summary statistic over time that can be embedded to explore population structure and variability statically, which we use to organise subjects in our test set by age. Then, after partitioning subjects into cohorts, we propose a novel method for producing a time-varying trajectory of persistence diagrams that can be used to quantify the progression and entropy of brain states. In summary, we make the following contributions:

- We present a novel non-parametric framework for transforming time-varying fMRI data into time-varying topological representations.
- We empirically show that these representations (i) capture age-related differences, and (ii) shed light on the cognitive processes of age-stratified cohorts.
- Finally, we show that our topological features are more informative for an age prediction task than other representations of the data set.

## 2   Background on topological data analysis

Topological data analysis (TDA) recently started gaining traction in machine learning [13, 32–35, 37, 40, 45, 47, 48, 50, 67]. TDA is a rapidly-growing field that provides tools for analysing the shape of data sets. This section provides a brief overview, aiming primarily for intuition and less for depth (see also Section A.1 for a worked example). We refer to Edelsbrunner and Harer [25] for

details. To our knowledge, this is the first time that TDA has been *directly* applied to fMRI data (as opposed to applying it on auxiliary representations such as functional connectivity networks).

**Simplicial homology.**   The central object in algebraic topology is a simplicial complex K, i.e. a high-dimensional generalisation of a graph, containing simplices of varying dimensions: vertices, edges, triangles, tetrahedra, and their higher-dimensional counterparts. A graph, for example, can be seen as a 1-dimensional simplicial complex, containing vertices and edges. Such complexes are primarily used to describe topological objects such as manifolds[1]. Simplicial homology refers to a framework for analysing the connectivity of K via matrix reduction algorithms, assigning K a graded set of mathematical groups, the homology groups. Homology groups describe the topological features of K; in low dimensions $d$, these features are called *connected components* ($d = 0$), *tunnels* ($d = 1$), and *voids* ($d = 2$), respectively. The number of $d$-dimensional topological features is referred to as the $d$th Betti number $\beta_d \in \mathbb{N}$; it is used to distinguish between different topological objects. For example, a circle (i.e. the *boundary* of a disk) has Betti numbers $(1, 1)$ because there is a single connected component and a single tunnel, while a filled square has Betti numbers $(1, 0)$.

**Persistent homology.**   The analysis of real-world data sets, having no preferred scale at which features occur, requires a different approach: Betti numbers cannot be directly used here because they only represent counts, i.e. a single scale. Endowing them with additional information leads to *persistent homology*, an extension of simplicial homology that requires a simplicial complex K and an additional function $f \colon \mathrm{K} \to \mathbb{R}$, such as an activation function. If $f$ only attains a finite set of function values $f_0 \le f_1 \le \cdots \le \ldots f_{m-1} \le f_m$, one can sort K according to them, leading to a *filtration*—a nested sequence of simplicial complexes

$$\emptyset = \mathrm{K}_0 \subseteq \mathrm{K}_1 \subseteq \cdots \subseteq \mathrm{K}_{m-1} \subseteq \mathrm{K}_m = \mathrm{K}, \tag{1}$$

with $\mathrm{K}_i := \{\sigma \in K \mid f(\sigma) \le f_i\}$. Filtrations represent the evolution of K along $f$. Similar to the Watershed transform in image processing [49], topological features can be *created* (a new connected component might arise) or *destroyed* (two connected components might merge into one) in a filtration. Persistent homology efficiently tracks topological features across a filtration, representing each one of them as a tuple $(f_i, f_j) \in \mathbb{R}^2$, with $i \le j$ and $f_i, f_j \in \mathrm{im}(f)$.

**Persistence diagrams.**   The tuples $(f_i, f_j)$ are collected according to their dimension $d$ and stored in the $d$th *persistence diagram* $\mathcal{D}_d$, which summarises all $d$-dimensional topological activity. As a consequence of the calculation process, all points in $\mathcal{D}_d$ are situated *above* the diagonal. The quantity $\mathrm{pers}(x, y) := |y - x|$, i.e. the distance to the diagonal (up to a constant factor), of a point $(x, y) \in \mathcal{D}_d$ is called the *persistence* of its corresponding topological feature. Low-persistence features used to be considered 'noise', while high-persistence features are assumed to correspond to 'real' features of a data set [26]. Recent work cast some doubts as to whether this assumption is justified [11]; in medical data, low persistence merely implies 'low reliability', *not* necessarily 'low importance.'

## 3   Related work

For fMRI analysis, the typical approach is to compare voxel activations directly, but when one is interested in time-varying activity from a continuous stimulus (e.g. while watching a movie or resting), voxel data is sometimes transformed into correlation matrices, either calculated across *time points* [6] or across *voxels* [65]. In the latter case, the goal is to study functional connectivity, i.e. information about the connectivity between brain regions sharing certain functional properties. Due to the size of the resulting matrices, one also often reduces the dimensionality by applying an atlas parcellation [54]. Both of these representations are efficacious, with voxel-by-voxel correlation matrices providing insights into the topology and dynamics of human brain networks [60]. Moreover, for many multi-subject fMRI studies, *shared response models* [15], abbreviated as SRMs, have proven effective. SRMs 'learn' a mapping of multiple subjects into the same space, enabling the detection of group differences, or the study of relations between brain activity and movie annotations, for example [62]. SRM was recently used to map voxel activity into a functional space (as opposed to an anatomical one) in order to study the brain representation of, among others, visual and auditory

information while receiving naturalistic audiovisual stimuli [36]. Nevertheless, while it is one of the most powerful techniques for extracting cognitively-relevant signals from fMRI data, there is still room for improvement.

Previous work fusing (f)MRI analysis and topological data analysis is either based on auxiliary (topological) representations [51, 56], such as the MAPPER algorithm [55] which operates on graphs, and requires numerous parameter choices, or it makes use of functional connectivity information (information about connectivity between brain regions sharing functional properties) and pre-defined regions of interest [5, 17, 27, 31, 52]. Some studies have investigated topological approaches on other measuring modalities, such as structural MRI for anatomical analyses [16], or diffusion MRI/DTI for studying white matter integrity [17]. By contrast, our method operates *directly* on fMRI volumes, requiring neither additional location information nor auxiliary representations. We will instead make use of *cubical complexes*, for which we essentially replace triangles by squares and tetrahedra by cubes (see Figure 1c and the subsequent section for details). Cubical complexes and their homology are well-studied in algebraic topology, but their use in real-world applications used to be limited to image segmentation [2]. This changed with the rise of persistent homology, which was extended to the cubical setting [44, 59, 63], leading to cubical persistent homology [23, 41, 64].

## 4  A topology-based framework for fMRI data sets

In the following, we will be dealing with time-varying fMRI. By this, we mean that we are observing an activation function $f: \mathcal{V} \times \mathcal{T} \to \mathbb{R}$ over a 3D bounded volume $\mathcal{V} \subset \mathbb{R}^3$ and a set of time steps $\mathcal{T}$. The alignment of $\mathcal{V}$ across different subjects is highly non-trivial; we provide more details about this at the beginning of Section 5. For $t \in \mathcal{T}$, the function $f(\cdot, t)$ is typically visualised using either stacks of images (Figure 1a) or volume rendering (Figure 1b). While it would be possible to analyse the topology of individual images [9], we want a holistic view of the topology of $\mathcal{V}$. To this end, we transform $\mathcal{V}$ into a *cubical complex* C, i.e. an equivalent of a simplicial complex, in which triangles and tetrahedra are replaced by squares and cubes (see Figure 1c). Cubical complexes are perfectly suited to represent an fMRI volume $\mathcal{V}$ because each voxel corresponds precisely to one cubical simplex (whereas if we were to use a simplicial complex, we would have to employ interpolation schemes as there is no natural mapping from voxels to tetrahedra; see Figure A.3 for more details).

**Terminology.**     We assume that we are given a data set of $n$ volumes $\mathcal{V}_1, \ldots, \mathcal{V}_n$, corresponding to $n$ different individuals, and a set of $m$ time steps $\mathcal{T} = \{t_1, \ldots, t_m\} \subset \mathbb{N}$. We use $\mathrm{vert}(\mathcal{V}_i)$ to denote the vertex (i.e. voxel) set of $\mathcal{V}_i$, and $f_i$ to denote its activation function, i.e. $f_i: \mathcal{V}_i \times \mathcal{T} \to \mathbb{R}$, Here, the activation functions are *aligned* with respect to their time steps; this is an assumption that greatly simplifies all subsequent analysis steps. It does not impose a large restriction in practice.

**Topological features from fMRI data.**     We obtain topological features of each $f_i$ following a three-step procedure, namely (1) cubical complex conversion, (2) filtration calculation, and (3) persistence diagram calculation. The *conversion* of a volume $\mathcal{V}_i$ to a cubical complex $C_i$ is simple, as $\mathcal{V}_i$ and $C_i$ share the same cubical elements and connectivities. Thus, the vertices of $C_i$ are the voxels of $\mathcal{V}_i$ and there are edges between neighbouring vertices as defined by a regular 3D grid, in which each vertex has six neighbours (two per coordinate axis). These neighbourhoods implicitly define the connectivity of higher-dimensional elements (squares and cubes). We will use $\sigma$ to denote an element of a cubical complex[2]. Next, we impose a *filtration*—an ordering—of the elements of $C_i$. Since we want to analyse topological features over time, we have to calculate one filtration for every time step. Given $t_j \in \mathcal{T}$, we assign the values of $f_i(\cdot, t_j)$ to $C_i$. We use the most natural assignment: each vertex (voxel) of $C_i$ receives its activation value at time $t_j$, while a higher-dimensional element $\sigma$ is assigned a value recursively via $f_i(\sigma, t_j) := \max_{v \in \mathrm{vert}(\sigma)} f_i(v, t_j)$. We then sort the cubical complex $C_i$ in ascending order according to these values; in case of a tie, a lower-dimensional element (e.g. an edge) precedes a higher-dimensional one (e.g. a square). Having obtained a filtration according to Equation 1, we may now calculate the persistent homology of $C_i$ at time step $t_j$, resulting in a collection of *persistence diagrams*. Since each $\mathcal{V}_i$ is three-dimensional, we obtain a triple $\left( \mathcal{D}_0^{(i,j)}, \mathcal{D}_1^{(i,j)}, \mathcal{D}_2^{(i,j)} \right)$ for every time step $t_j$; persistence diagrams for $d \geq 3$ are all empty. Notice that the calculation of persistence diagrams for a participant $i$ and a time step $t_j$ can be easily

parallelised since we treat time steps independently. Subsequently, we will use $\mathcal{D}^{(i)}$ to denote the set of all persistence diagrams associated with the $i$th participant. We can plot the resulting persistence diagrams of each participant as a set of diagrams in $\mathbb{R}^3$, with the additional axis being used to represent *time* (Figure 1d).

The filtration that we employ here is also known as a *sublevel set filtration*. Other filtrations [3] could also be used (our method is not restricted to any specific one), but a symmetry theorem [21] states that unless we are willing to modify the activation function values themselves we are not gaining any more information about the topology of our input data. For the subsequent analyses, we will be dealing with collections of persistence diagrams $\mathcal{D}^{(i)}$. The space of persistence diagrams affords several metrics [20, 22], but they are computationally expensive and infeasible for the cardinalities we are dealing with (a typical persistence diagram of a participant contains about 10,000 features). We will thus be working with *topological summary statistics* and *persistence diagram vectorisations*.

**Properties.**   Prior to delving deeper into our pipeline, we describe some properties of our approach and why topological features are advantageous. Topology is inherently coordinate-free, meaning that all the features we describe are invariant to homeomorphism, i.e. stretching and bending. Moreover, the persistence diagrams of spaces of different cardinalities and scales can be compared, making it possible to 'mix' participants from studies with different imaging modalities or resolutions (of course, this should not be done indiscriminately). Arguably the largest advantage of persistent homology is its stability with respect to perturbations [20, 22, 57]. This is quantified by the following theorem, whose proof we defer to Section A.6.

**Theorem 1.** *Let $f\colon \mathcal{V} \to \mathbb{R}$ and $g\colon \mathcal{V} \to \mathbb{R}$ be two activation functions. Then their corresponding persistence diagrams $\mathcal{D}_f$ and $\mathcal{D}_g$ satisfy $\mathrm{W}_\infty(\mathcal{D}_f, \mathcal{D}_g) \leq \|f - g\|_\infty$, where $\mathrm{W}_\infty$ denotes the bottleneck distance between persistence diagrams, defined as $\mathrm{W}_\infty(\mathcal{D}_f, \mathcal{D}_g) := \inf_{\eta\colon \mathcal{D} \to \mathcal{D}_g} \sup_{x \in \mathcal{D}_f} \|x - \eta(x)\|_\infty$, with $\eta\colon \mathcal{D}_f \to \mathcal{D}_g$ denoting a bijection between the points of the two diagrams, and $\|\cdot\|_\infty$ referring to the $\mathrm{L}_\infty$ norm.*

The consequence of this stability theorem is that the persistence diagrams that we calculate are stable with respect to perturbations, provided those perturbations are of small amplitudes. This is a desirable characteristic for a feature descriptor because it provides us with well-defined bounds for its behaviour under noise. A more precise version of this stability theorem exists [22], but requires a more involved setup[3], which we leave for future work. In general, we note that time-varying TDA is still a rather nascent sub-field of TDA. A standard approach, namely the calculation of 'persistence vineyards' [19], resulting in a decomposition of a time-varying persistence diagram into individual 'vines', is not applicable here because the changes between different time steps are not infinitesimal (there *is* a large amount of temporal coherence between consecutive time steps, but there is no guarantee that the change between them is upper-bounded). It is still unknown in our setting whether a vineyard representation with unique vines exists at all [42]. We therefore prefer to treat the individual time steps as independent calculations but note that future work should address a more efficient computation by exploiting similarities between consecutive time steps.

**Implementation and complexity.**   While the general calculation of persistent homology on high-dimensional simplicial complexes is still computationally expensive, there are highly-efficient algorithms for lower-dimensional calculations [8]. Here, we use DIPHA [7], a distributed implementation of persistent homology, as it implements an efficient algorithm for computing topological features of cubical complexes [63]. Space and time complexity is linear in the number of voxels, so our conversion process does not change the complexity of processing the data. The persistent homology calculation has a time complexity of $\mathcal{O}(|\mathcal{V}|)^\omega$, with $\omega \approx 2.376$ [38]. The distributed implementation of DIPHA is reported [7] to be capable of calculating persistent homology for $|\mathcal{V}| \approx 10^9$, making our pipeline feasible and scalable. For the persistence image calculation in Section 5.2, we use Scikit-TDA [53]. We make our code publicly available[4] to ensure reproducibility.

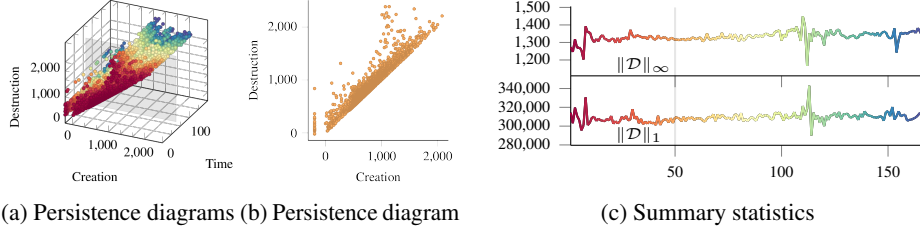

(a) Persistence diagrams (b) Persistence diagram        (c) Summary statistics

Figure 2: Example of summary statistics calculations. Starting from a sequence of time-varying persistence diagrams (a) of one participant, for each diagram slice (b), we evaluate a scalar-valued statistic $S\colon \mathcal{D} \to \mathbb{R}$, leading to a time series (c); the corresponding time point is highlighted.

## 5 Results

We evaluate our topological pipeline using open-source fMRI data [46], available on the OpenNeuro database (accession number ds000228). The participants comprised 33 adults (18–39 years old; M = 24.8, SD = 5.3; 20 female) and 122 children (3.5–12 years old; M = 6.7, SD = 2.3; 64 female) who watched the same animated movie 'Partly Cloudy' [58] while undergoing fMRI. Please refer to Section A.3 and Yates et al. [66] for a full description of the pre-processing. The relevant outputs of these pre-processing steps are: a 4-dimensional ($x \times y \times z \times t$, with $x, y, z$ being coordinates, and $t$ representing time) fMRI time series and a whole-brain mask (BM) for each individual subject. The 4D volume of each participant has dimensions $65 \times 77 \times 60 \times 168$. Each of 168 time steps of the fMRI time series comprises 2 s of the movie and corresponds to the same point in the movie for each subject; since for the first five time steps only a blank screen was shown, we remove these plus two time steps to account for the fMRI hemodynamic lag for all analyses. We supplemented the whole-brain mask by also creating an 'occipital-temporal' mask (OM) for each subject. This entailed finding the intersection between an individual subject's whole-brain mask and occipital, temporal, and precuneus regions of interest defined from the Harvard–Oxford cortical atlas. If our results reflect patterns relevant to cognitive processing, we would expect similar—if not better— results using this occipital-temporal mask, since it contains the regions most consistently involved in movie-watching (e.g. visual regions). Last, we also calculated the 'logical XOR' between the whole-brain mask and the occipital-temporal mask; this mask (XM) makes it possible to study the relevance of topological features with respect to non-visual regions (including the frontal lobe) in the brain. To prevent analysis bias, data were initially fully unlabelled during the development of our pipeline. Later on, participants were assigned to cohorts based on their age group, using the same bins as Yates et al. [66]; we initially did not know whether cohorts were sorted in ascending or descending order. The actual ages were only used in the age prediction experiment, which was performed *after* method development had ceased.

### 5.1 Static analysis based on summary statistics

Extracting information from the time-varying persistence diagrams of each participant is impeded by their complex geometrical structure, making it necessary to use summary statistics. We first focus on a description of *global* properties of participants, restricting ourselves to persistence diagrams with $d = 2$ (i.e. we are studying voids of the activation function). To this end, we calculate topological summary statistics of the form $S\colon \mathcal{D} \to \mathbb{R}$. We calculate two related summary statistics here, namely the *infinity norm* $\|\mathcal{D}\|_\infty$ of a persistence diagram [20] and the *$p$-norm*[5] $\|\mathcal{D}\|_p$ [14, 22], defined by

$$\|\mathcal{D}\|_\infty := \max_{x,y\in\mathcal{D}} \mathrm{pers}(x,y)^p \quad \text{and} \quad \|\mathcal{D}\|_p := \sqrt[p]{\sum_{x,y\in\mathcal{D}} \mathrm{pers}(x,y)^p}, \tag{2}$$

with $p \in \mathbb{R}$. We found $p = 1$ to be sufficient, thus using unscaled persistence values. Since both norms in Equation 2 yield one scalar value for a persistence diagram, the summary statistics turn a sequence of time-varying persistence diagrams into a time series of scalar-valued summary statistics. Figure 2 depicts this for a single participant from our data (for illustrative purposes, we show *all* 168 time steps; as specified before, only 161 time steps will be used for the subsequent analyses).

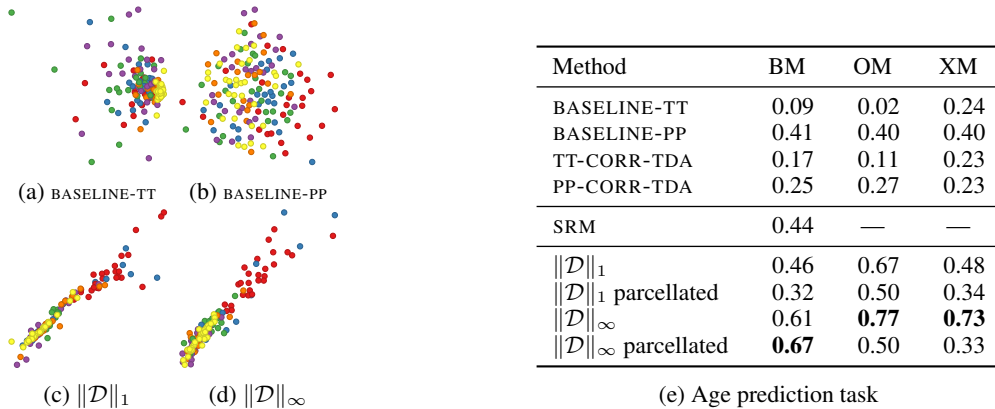

|                          | (a) BASELINE-TT | (b) BASELINE-PP |  |
|--------------------------|-----------------|-----------------|--|

| Method                    | BM   | OM   | XM   |
|---------------------------|------|------|------|
| BASELINE-TT               | 0.09 | 0.02 | 0.24 |
| BASELINE-PP               | 0.41 | 0.40 | 0.40 |
| TT-CORR-TDA               | 0.17 | 0.11 | 0.23 |
| PP-CORR-TDA               | 0.25 | 0.27 | 0.23 |
| SRM                       | 0.44 | —    | —    |
| $\|\mathcal{D}\|_1$                 | 0.46 | 0.67 | 0.48 |
| $\|\mathcal{D}\|_1$ parcellated     | 0.32 | 0.50 | 0.34 |
| $\|\mathcal{D}\|_\infty$                 | 0.61 | **0.77** | **0.73** |
| $\|\mathcal{D}\|_\infty$ parcellated     | **0.67** | 0.50 | 0.33 |

(c) $\|\mathcal{D}\|_1$      (d) $\|\mathcal{D}\|_\infty$        (e) Age prediction task

Figure 3: An embedding of the *distances* for different baselines and topological summaries, based on the whole-brain mask (BM); colour-coding refers to the age group of participants. The table depicts the results of the age prediction task, stratified by different brain masks; performance is measured as a correlation coefficient (bold indicates the best results).

**Qualitative evaluation.** Figure 3 shows an embedding obtained from our topological summary statistics (using multidimensional scaling based on the Euclidean distance between per-participant curves) compared to baseline embeddings, which we obtain from the two correlation matrices described in Section 3. We refer to them as BASELINE-TT (time-based) and BASELINE-PP (voxel-based; parcellated for computational ease), respectively (see Section A.4 for additional details). Both topology-based embeddings are showing a split between participants. By colour-coding the age group of each participant, we see that topology-based embeddings separate adults (red) from children (other colours). The baselines, by contrast, do not exhibit such a clear-cut distinction.

**Quantitative evaluation.** To *quantify* the benefits of our proposed topological feature extraction pipeline, we set up a task in which we predict the age of the non-adult participants. Using a ridge regression and leave-one-out cross-validation (see Section A.4 for detailed descriptions of all comparison partners and Section A.5 for additional experimental details), we train models on either the curves of summary statistics (not the embeddings) the baseline matrices, and additional topological baselines, reporting the correlation coefficient in the table in Figure 3. Higher values indicate that the model is better suited to predict the age. The SRM result comes from previous work on the same data set [66]; we note that our task is slightly different[6]. Overall, we observe strong correlations, indicating that topological features are highly useful for age prediction and carry salient information. Performance based on the occipital-temporal mask (OM) and on the XOR mask (XM) is higher than for the whole-brain mask (BM); we hypothesise that this is partially due to the higher noise level of BM, whereas OM and XM focus only on a subset of the brain (which decreases the noise level). We also note that $\|\mathcal{D}\|_\infty$, which only considers the most persistence topological feature of a persistence diagram, performs best in the prediction task, possibly because it is more robust to small-scale noise. Interestingly, parcellated data (i.e. highly coarse representations) applied to our cubical complex filtration outperforms the whole-brain mask. This is the *only* one of the parcellated volumes to do so. We speculate that the coarsening helps to remove some noise here, whereas the other masks, containing fewer voxels, are less noisy by construction and contain more fine-grained information that is suppressed by the coarsening.

## 5.2 Dynamic analysis based on brain state trajectories

So far, we dealt only with overall summary statistics. Our framework also enables analysing the brain state of participants over time. We sidestep the aforementioned issue of persistence diagram metric computations by calculating *persistence images* [1] from the persistence diagrams. A persistence image is a function $\Psi\colon \mathbb{R}^2 \to \mathbb{R}$ that turns a diagram $\mathcal{D}$ into a surface via $\Psi(z) := \sum_{x,y\in\mathcal{D}} \mathrm{w}(x,y)\Phi(x,y,z)$, where $\mathrm{w}(\cdot)$ is a fixed piecewise linear weight function and $\Phi(\cdot)$ denotes a probability distribution, which is typically chosen to be a normalised symmetric Gaussian. By

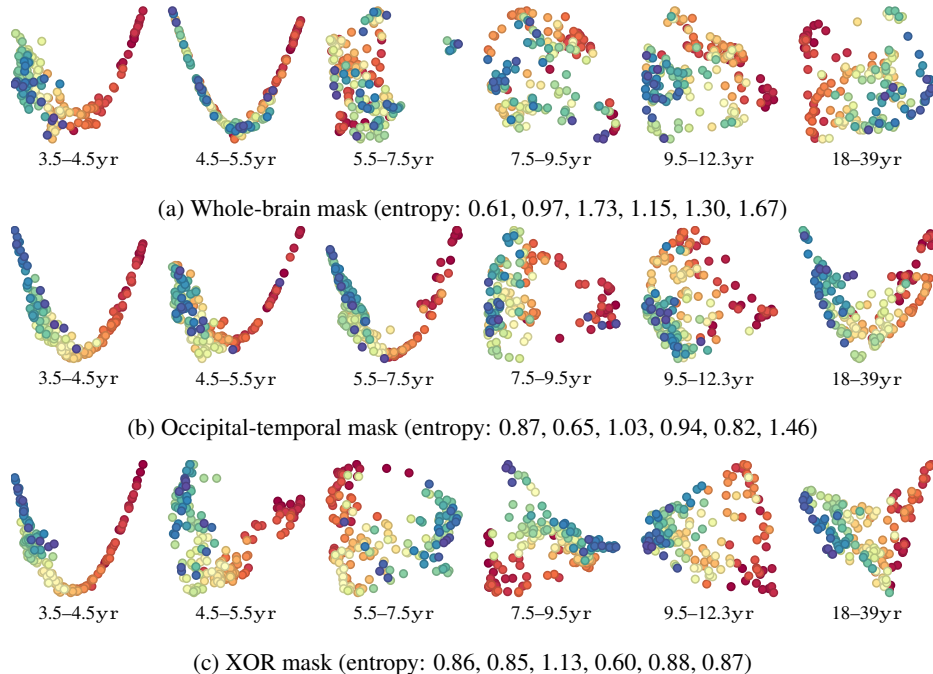

(a) Whole-brain mask (entropy: 0.61, 0.97, 1.73, 1.15, 1.30, 1.67)

(b) Occipital-temporal mask (entropy: 0.87, 0.65, 1.03, 0.94, 0.82, 1.46)

(c) XOR mask (entropy: 0.86, 0.85, 1.13, 0.60, 0.88, 0.87)

Figure 4: Cohort brain state trajectories for different brain masks, embedded using PHATE [39]. Annotations provide the age range of subjects in one cohort. We also report the von Neumann entropy of the respective diffusion operator [4].

discretising $\Psi$ (using an $r \times r$ grid), a persistence diagram is transformed into an image[7]; this is depicted in Figure 1e. The main advantage of $\Psi$ lies in embedding persistence diagrams into a space that is amenable to standard machine learning tools; moreover, $\Psi$ affords defining and calculating *unique* means, as opposed to persistence diagrams [42, 43, 61]. Subsequently, we use $r = 20$ and a Gaussian kernel with $\sigma = 1.0$; $\Psi$ is known to be impervious to such choices [1].

### 5.2.1 Cohort brain state trajectories

By evaluating $\Psi(\mathcal{D}_2^{(i,j)})$ for each time step $t_j$, we turn the sequence of persistence diagrams of the $i$th participant into a matrix $\mathbf{X}^{(i)} \in \mathbb{R}^{m \times r^2}$, where the $j$th row corresponds to the 'unravelled' persistence image of time step $t_j$. We now calculate the sample mean $\overline{\mathbf{X}}_k$ of each participant cohort, resulting in six matrices whose rows represent the average topological activity of participants in the respective cohort. Taking the Euclidean distance between persistence images as a proxy for their actual topological dissimilarity [1, Theorem 3], we calculate pairwise distances between rows of each $\overline{\mathbf{X}}_k$ and embed them using PHATE [39], a powerful embedding algorithm for time-varying data. This turns $\overline{\mathbf{X}}_k$ into a 2D *brain state trajectory* (where the state is measured using topological features). Figure 4 depicts the resulting trajectories for different masks. All brain state trajectories exhibit visually distinct behaviour in older and younger subjects. The youngest subjects are characterised by a simple 'linear' trajectory in the whole-brain mask, indicating that their processing of the movie is more sensory-driven. This pattern is visible in Figure 4b for young children in general: until 7.5 yr, sensory processing, analysed using the occipital-temporal mask, is comparatively simple. In older subjects, we observe more complex trajectories with higher entropy generally. Developmental differences are best indicated in Figure 4c, where we see that the overall trajectory shape becomes 'adult-like' *earlier* (and thus more complex). Since this mask is composed of more cognitive brain regions (rather than sensory ones), we hypothesise that this could indicate that older participants, including older children, are capable of connecting different aspects of the movie to their memories, for example, whereas the simpler trajectories of the two youngest cohorts in *all* brain masks may indicate that these participants are not comprehending the movie on a non-superficial level.

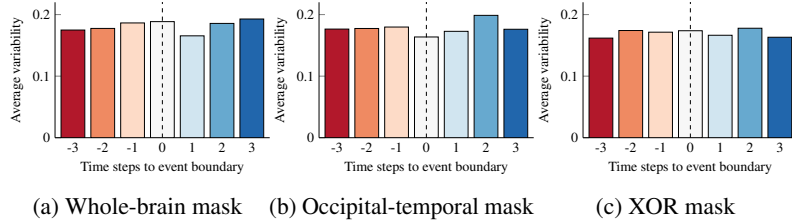

(a) Whole-brain mask     (b) Occipital-temporal mask     (c) XOR mask

Figure 5: Histograms showing the mean across-cohort variability as a function of the distance to an event. The $x$-axis shows the time steps prior to (negative) or after (positive) an event boundary, while their $y$-axis depicts across-cohort variability. Please refer to Section 5.2.2 for more details.

### 5.2.2 Variability analysis

To quantify the variability across cohorts, we calculate the per-column maximum of each $\overline{\mathbf{X}}_k$, referring to the respective set of values as $\|\overline{\mathbf{X}}_k\|_\infty \in \mathbb{R}^m$; the calculated values are the equivalent of the infinity norm evaluated (per time step) on a mean persistence image of the cohort. We finally calculate $s\left(\|\overline{\mathbf{X}}_1\|_\infty, \dots, \|\overline{\mathbf{X}}_6\|_\infty\right)$, i.e. the sample standard deviation per time point, thus obtaining a *variability curve* of $m$ time steps (see Figure A.5). To use this variability curve, we ran an online study to discover which salient events are detected by participants in the movie. Using 22 test subjects (with no overlap to the ones used in the fMRI data acquisition process), we followed Ben-Yakov and Henson [10] and determined consensus boundaries of events in the movie. We declare an event boundary to be *salient* if at least 7 participants agree, resulting in 20 events. Given this information, we collect the average variability over all events for a window of $w = 3$ time steps before and after an event, leading to averaged variabilities $\{s_1, \dots, s_7\}$, where $s_4$ corresponds to the average variability at the event boundary itself (see Figure 5). It is our hypothesis that post-event and pre-event variability are different—in other words, our topological features capture cognitive differences across cohorts and events. To quantify this, we calculate $s_{\text{pre}} := \max_{i \leq 3} s_i - \min_{i \leq 3} s_i$ and $s_{\text{post}} := \max_{i \geq 5} s_i - \min_{i \geq 5} s_i$. We set $\theta := s_{\text{pre}} - s_{\text{post}}$ as our test statistic and perform a bootstrap procedure by sampling 20 time points at random and repeating the same calculation, thereby obtaining an empirical null distribution. This results in bootstrap samples $\widehat{\theta}_1, \dots, \widehat{\theta}_{1000}$ serving as a null distribution $\widehat{\theta}$, from which we obtain the achieved significance level (ASL) as $\Pr(\widehat{\theta} \geq \theta)$.

The ASL values are $0.084$ (whole-brain mask, BM), $0.045$ (occipital-temporal mask, OM), and $0.396$ (XOR mask, XM), respectively, indicating that the effect of capturing events is *strongest* in OM and significant at the $\alpha = 0.05$ level. This aligns well with the gradual differences between cohorts expressed in Figure 4b. Event differences are less pronounced in BM (which, as Figure 4a shows, is capturing more complex cohort patterns). Finally, event differences are *absent* in XM, showing that across-cohort variability is not consistent with event boundaries here, hinting at the fact that this mask might better be used to assess within-cohort variability rather than across-cohort variability. Please refer to Section A.7 for additional visualisations.

## 6 Conclusion

This paper demonstrates the potential of an unsupervised, non-parametric topology-based feature extraction framework for fMRI data, permitting both static and dynamic analyses. We showed that topological summary statistics are useful in an age prediction task. Using vectorised topological features descriptors, we also developed cohort brain state trajectories that show the time-varying behaviour of a cohort of participants (binned by age). Next, to highlight qualitative age-related differences in the overall cognition of participants, we were also able to uncover quantitative differences in event processing. In the future, we want to further analyse the *geometry* of brain state trajectories and link states back to events; a preliminary analysis (see Section A.8) finds significant differences between the mean curvature [24] of adult and non-adult participants, thus showcasing the explanatory potential of topological features. We also plan on investigating geometrical aspects of topological features [29, 68] as well as their large-scale validation based on synthetic data generators [28].

## Broader impact

The primary contribution of this work—a novel, parameter-free way of extracting informative features from fMRI data—is of a computational nature. In general, we fully acknowledge that any researcher dealing with fMRI data analysis (not necessarily restricted to machine learning methods) has a big responsibility. Since our work is purely computational, we do not believe that it will have adverse ethical consequences, provided the experimental design is unbiased. For the same reason, our work is not specifically favouring or disfavouring any groups.

Beyond the immediate applications for fMRI data analysis, our work also has a broader applicability for the analysis of time-varying or structured neuroscience data in general. This includes other non-invasive techniques such as EEG or MEG, but also neuronal spike data from cell populations. Our work is appealing for such data because it does not *require* auxiliary representations such as graphs. We are thus convinced that the introduction of our directly-computable topological features will overall have beneficial outcomes.

As long-term goal, for example, our work could serve as a foundation to investigate neurological pathologies (such as depressive disorders) from a new, topological perspective. In general, our dynamic analyses also allow us to capture not just stable traits in different populations, but also the different mental states participants progress through while undergoing fMRI. As a generic feature descriptor of brain states, we would welcome a future in which topological features aid in understanding such traits or states.

## Acknowledgments and Disclosure of Funding

We thank the anonymous reviewers for their valuable comments, which helped us improve the paper. The first author is also indebted to Michael Moor, Leslie O'Bray, and Caroline Weis for their constructive feedback in preparing the manuscript. This work was partially funded and supported by the Swiss National Science Foundation [Spark grant 190466, *B.R.*], the Alfried Krupp Prize for Young University Teachers of the Alfried Krupp von Bohlen und Halbach-Stiftung [*K.B.*], IVADO Professor startup & operational funds [*G.W.*], an NSF Graduate Research Fellowship [*T.Y.*], NSF grant CCF 1839308 & Canadian Institute for Advanced Research [*N.T.B.*], Chan-Zuckerberg Initiative grants 182702 & CZF2019-002440 [*S.K.*], and NIH grants R01GM135929 & R01GM130847 [*G.W., S.K.*]. The content provided here is solely the responsibility of the authors and does not necessarily represent the official views of the funding agencies. The funders had no role in study design, data collection & analysis, decision to publish, or preparation of the manuscript.

## Footnotes

[1]We will deviate from this notion later on in this paper but follow the conventional exposition for now, which focuses primarily on a simplicial view.

[2]These elements are the 'simplices' of the cubical complex, but we refrain from re-using the term 'simplex' so as not to confuse ourselves or the reader.

[3]We will have to show Lipschitz continuity for the functions, plus certain other properties of the space $\mathcal{V}$.

[4]https://github.com/BorgwardtLab/fMRI_Cubical_Persistence

[5]The term *total persistence* is sometimes used interchangeably for this norm.

[6]Yates et al. [66] learn a shared set of features in adult participants to predict the age of non-adults.

[7]Intuitively, this can also be seen as a form of kernel density estimation on persistence diagrams.

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
