[Supplementary Material]

# A  Appendix

The following sections provide additional details about the experiments as well as a brief glimpse into other analyses that we are actively pursuing for future work.

## A.1  An intuitive introduction to persistent homology

Persistent homology was developed as a 'shape descriptor' for real-world data sets, where the idealised notions of algebraic topology do not necessarily apply any more. This is illustrated by the subsequent figure, which deals with a point cloud that has a roughly circular shape. Notice that this shape is immediately recognisable to humans, but from the perspective of algebraic topology, it is merely a collection of points with a trivial shape.

We observe that we *can* analyse this point cloud by picking an appropriate scale parameter. More precisely, if we start connecting points that are within a certain distance $\epsilon$ to each other, we obtain a nested sequence of simplicial complexes (in our context, this term is synonymous with a graph) as we increase $\epsilon$. This is a special type of filtration—a filtration based on pairwise distances, and the resulting simplicial complexes are depicted above. It is now possible to calculate Betti numbers for each of these complexes. Since we are only dealing with 2D points, there are only two relevant Betti numbers, namely $\beta_0$ and $\beta_1$, corresponding to the number of connected components and the number of cycles, respectively. Suppose now that we *track* these numbers for each one of the steps in the filtration; moreover, suppose we have a way of making the individual steps in the filtration as small as possible such that we never miss any changes in $\beta_0$ and $\beta_1$. For every topological feature—every component and every cycle—we can thus measure precisely when a feature was created and when it was destroyed.

**Persistence diagram.**  This information is collected in the persistence diagram, which summarises all topological activity. In this example, the persistence diagram of the 1-dimensional topological features contains a few points, each one of them corresponding to one specific cycle in the data. The axes correspond to the scale parameter (their actual values can be safely ignored for this illustrative example). The $x$-axis shows the threshold at which a cycle was *created*, i.e. at which there is a 'hole' in the corresponding simplicial complex, while the $y$-axis depicts the threshold at which this hole is *destroyed*, i.e. closed. We do not specifically indicate this here, but cycles are destroyed whenever all points that are involved in their creation are connected to each other. Put differently, this means that we ignore cycles created by individual triangles of points, for example, as they are qualitatively different from cycles created by arranging points in such a circular shape (there are more technical reasons for this restriction). In any case, the persistence diagram demonstrates that virtually all cycles—depicted as points—occur at small scales, except for *one*. This coincides with our intuition: we do not perceive such a point cloud to have a lot of large-scale cycles. The persistence diagram thus serves as an intuitive feature descriptor: points that occur at large scales are far removed from the diagonal (and have a high persistence), whereas the small-scale features cluster around the diagonal.

A persistence diagram of the 1-dimensional topological features (cycles).

The interesting fact is that knowing the persistence diagram also makes it possible for us to 'guess' the number of relevant scales of a point cloud! In this example, we would possibly state that there is only *one* useful scale at which to analyse the data, namely the scale for which the cycle structure becomes topologically apparent. In general, this will differ based on the data set. Persistent homology does not force us to prefer a scale, making it suitable for the analysis of real-world data sets. The ingenious realisation of Edelsbrunner et al. [26] was that there is no reason to 'guess' scales or compute Betti

numbers per step, as we described it above. Instead, it is possible to obtain information about *all* potential scales by a single pass through the data, making this a highly-efficient algorithm (at least as long as the dimension of the input data is bounded; calculating topological features for dimensions $d \gg 3$ efficiently is still a topic of ongoing research).

**Persistence images.** Since the metric structure of persistence diagrams is known to be complex [43], various kernel-based and 'vectorisation' methods exist. In the main text, we focus on *persistence images* [1], a technique that essentially estimates the density of a persistence diagram and uses a grid to obtain a fixed-size representation. Such representations may then be used for downstream processing tasks. As a worked example, consider the following persistence diagram. After rotating it so that the diagonal becomes the new $x$-axis, we can perform density estimates with different resolutions. The density estimator, which is by default a Gaussian kernel, can be adjusted as well, but Adams et al. [1] mention that this does *not* have a large influence on the results (whereas the resolution should be sufficiently large to capture differences). In the main paper, we use a smoothing value of $\sigma = 1$ and a resolution of $r = 20$, resulting in 400-dimensional vectors. We also calculated different resolutions and smoothing values, but the results are virtually identical, unless the resolution is decreased too much: recall that a single persistence diagram of one participant has around 10,000 features; reducing them to a, say, $5 \times 5$ image results in a large loss of information.

## A.2 Properties of cubical complexes

Figure A.3 depicts the differences between cubical complexes and simplicial complexes. The cubical complex is 'aligned' with a regular grid and does *not* force us to choose between an interpolation scheme. For a simplicial complex, however, the calculation of topological features in dimensions 1 and 2 necessitates the creation of 2-simplices, i.e. triangles. This, in turn, requires us to 'pick' between two triangulation schemes that result in different connectivities between the original vertices. In the worst case, this could lead to subtle differences in filtrations, since the new edges need to be weighted accordingly.

## A.3 fMRI pre-processing

The fMRI data acquisition used the following parameters: gradient-echo EPI sequence: TR = 2 s, TE = 30 ms, flip angle = 90°, matrix = $64 \times 64$, slices = 32, and interleaved slice acquisition. Data were collected using the standard Siemens 32-channel head coil for adults and older children. One of two custom 32-channel phased-array head coils was used for younger children (smallest coil: N = 3; M = 3.91, SD = 0.42 years old; smaller coil: N = 28; M = 4.07, SD = 0.42, years old). Acquisition parameters differed slightly across participants but all fMRI data were re-sampled to have the same voxel size, namely 3 mm isotropic with 10% slice gap. A T1-weighted structural image was also collected for all subjects (MPRAGE sequence: GRAPPA = 3, slices = 176, resolution = 1 mm isotropic, adult coil FOV = 256 mm, child coils FOV = 192 mm). Imaging data were pre-processed using `fMRIPrep` v1.1.8 [30].

## A.4 Baselines

As additional comparison partners, we calculate a time point correlation matrix and a spatial correlation matrix (see Section 3). These matrices are calculated from the time-varying fMRI data of a single participant, which is a 4D tensor indexed by time steps and spatial coordinates. By 'unravelling' the spatial dimensions of the tensor (using a *row-major ordering*, for example), the 4D tensor becomes a 2D tensor, i.e. a matrix in which each row corresponds to a single time step, and the columns correspond to voxels in the aforementioned order. From this $m \times N$ matrix, where $m$ denotes the

Figure A.3: If we use a cubical complex (left; only a single square cell is shown) we do not have to choose an interpolation scheme for voxel-based data. Function values can be stored in the vertices ($a$, $b$, $c$, $d$) and interpolation happens along the edges. For simplicial complexes, however, we need to convert the square into a triangle (the same issue occurs in higher dimensions with cubes and tetrahedra, respectively). This conversion to triangles leaves us with two ways of interpolating that will typically lead to different results. In one case, we are interpolating between $a$ and $c$, in the other case between $b$ and $d$. Neither one of these edges exist in the original data, though.

Figure A.4: A schematic illustration of the parcels used to make the computation of a full correlation matrix computationally feasible. This only pertains to the BASELINE-PP method.

number of time steps as in the main paper and $N$ denotes the total number of voxels, we can calculate Pearson product-moment correlation coefficients. If we do this for the original matrix, we obtain an $m \times m$ *time point correlation matrix*, which we denote by BASELINE-TT (i.e. a time-by-time matrix).

Conversely, we can transpose the matrix to obtain a voxel-by-voxel correlation matrix—referred to as a *full correlation matrix* (FCMA). This matrix has dimensions $N \times N$, though, which is computationally prohibitive for most applications. As a more feasible calculation, we calculate the spatial correlation matrix from a parcellated data set. We use 100 parcels from 17 functional networks Schaefer et al. [54], depicted in Figure A.4, so that we obtain a $100 \times 100$ correlation matrix, which we refer to as BASELINE-PP to indicate that parcellated data was used to obtain this matrix.

As additional comparison partners, we follow a more conventional topological data analysis pipeline and calculate persistence images from a set of correlation matrices. The matrix is treated as the adjacency matrix of a fully-connected graph, and we use a filtration that is specifically geared towards the analysis of such 'correlation graphs' [18]. Following our own pipeline, we convert the resulting persistence diagrams into persistence images (using smoothing values $\sigma \in \{0.1, 1.0\}$ and resolutions $r \in \{10, 20\}$, respectively), and report the best performance for the age prediction task. We denote the corresponding methods by TT-CORR-TDA and PP-CORR-TDA, depending on which correlation matrix was used for the calculation.

Last, as an ablation study, we use parcellated data with the same parcels as above and assign the respective values to the original cubical volume; each voxel corresponding to the same parcel is assigned the same value. This has the effect of coarsening information but also removing noise; while not decreasing the number of voxels in the data, it will decrease the number of topological features that have to be considered. We mark the results obtained using this technique with PARCELLATED.

### A.5 Age prediction experimental details

For the age prediction experiment from Section 5.1, we use a ridge regression classifier with internal leave-one-out cross-validation [12] for its regularisation strength parameter $C \in \{0.1, 1.0, 10.0\}$. Internally, the classifier optimises the $R^2$ score, i.e. the coefficient of determination. We report the correlation coefficient in the table in order to be aligned with the reporting in previous publications, though. We standardise *all* features prior to using them for the classifier. The classifier is then used in a leave-one-out cross-validation scheme.

Since the input features have different cardinalities (the parcellated voxel-by-voxel matrix, for example, has a cardinality of $100^2$, which will lead to severe overfitting, we reduce the baseline matrices to 100 dimensions using principal component analysis. Moreover, to demonstrate the impact of our topological summary statistics, we only use summary statistics from the second half of the

Table A.1: Additional experimental results for the age prediction tasks. In contrast to the table in the main paper, here we show both the correlation coefficient (CC; higher values are preferable ↑) and the mean squared error (MSE; lower values are preferable) whenever available.

| Method | BM | | OM | | XM | |
|---|---|---|---|---|---|---|
| | CC ↑ | MSE ↓ | CC ↑ | MSE ↓ | CC ↑ | MSE ↓ |
| BASELINE-TT | 0.09 | 10.15 | 0.02 | 13.81 | 0.24 | 7.19 |
| BASELINE-PP | 0.41 | 6.23 | 0.40 | 6.40 | 0.40 | 6.65 |
| TT-CORR-TDA | 0.17 | 10.04 | 0.11 | 12.57 | 0.23 | 9.76 |
| PP-CORR-TDA | 0.25 | 10.34 | 0.27 | 9.68 | 0.23 | 9.94 |
| SRM | 0.44 | 6.05 | — | — | — | — |
| $\|\mathcal{D}\|_1$ | 0.46 | 4.27 | 0.67 | 2.95 | 0.48 | 4.17 |
| $\|\mathcal{D}\|_1$ parcellated | 0.32 | 4.91 | 0.50 | 4.06 | 0.34 | 4.76 |
| $\|\mathcal{D}\|_\infty$ | 0.61 | 3.38 | **0.77** | **2.20** | **0.73** | **2.53** |
| $\|\mathcal{D}\|_\infty$ parcellated | **0.67** | **2.99** | 0.50 | 4.04 | 0.33 | 4.81 |

time series, resulting in *less* than 100 features. We observe that the results are highly stable; even reducing the number of selected features to less than 10 has no noticeable effect on the resulting regression model, indicating the informativeness of topology for this task.

Table A.1 shows extended results for this experiment, including MSE values (another goodness-of-fit measure) that were excluded from the table in the main paper because the incomparability to existing methods. To reproduce the values in this table, please use the provided `predict_age.py` script.

## A.6 Proof of the stability theorem

*Proof.* Let $\mathcal{V} := [0, 1]^3$; this is not a restriction because fMRI volumes are bounded, so they are always homeomorphic to this 'standard cube'. Hence, $\mathcal{V}$ is a compact metric space that can be triangulated. Since $f$ and $g$ are continuous functions (at least this is the 'idealised' view in which we have access to an infinite number of samples), the stability follows from the main theorem of Cohen-Steiner et al. [20]. ∎

## A.7 Across-cohort variability analysis

For the across-cohort variability analysis, Figure A.5 shows the 'raw' curves for each of the masks, annotated with the respective events. As described in Section 5.2.2, we *pool* variability for all events and analyse the resulting histograms. This construction loses some information, but is a simple way to assess overall differences in variability.

## A.8 Curvature analysis

The curvature $\kappa$ of a differentiable curve measures how sharply the trajectory curves at a given point *on* the curve. A circle, for example, always has curvature of 1 at each point, while a straight line, by contrast, always has a curvature of 0. We hypothesise that the curvature of brain state trajectories can help to further characterise subjects by looking at topological activity from a geometric point of view.

Let $x(t)$ be the $x$-coordinate of a brain trajectory (as shown in Figure 4) at time $t$ and $y(t)$ its respective $y$-coordinate. Furthermore, let us define $\dot{x}$ as the first derivative of $x$ with respect to $t$; equivalently, $\ddot{x}$ denotes the second derivative. Curvature can then be expressed as

$$\kappa = \frac{|\dot{x}\ddot{y} - \dot{y}\ddot{x}|}{(\dot{x}^2 + \dot{y}^2)^{\frac{3}{2}}} \tag{3}$$

Notice that curvature is an inherently *local* quantity. We computed $\kappa$ for all brain state trajectories of all cohorts and for all three segmentation masks. We then investigated the differences in $\kappa$ around event boundaries, similar to the variability analysis in Section 5.2.2. Figure A.6 shows the distribution of curvature values when stratifying the subjects into *adults* and *non-adults*. We find significant

(a) Whole-brain mask

(b) Occipital-temporal mask

(c) XOR mask

Figure A.5: Across-cohort variability curves for the different masks. The dotted lines represent the events. Generally, events are aligned with local extrema of the curves.

Figure A.6: Distribution of brain state trajectory curvature values at event boundaries. The distributions differ significantly for BM ($p_{\mathrm{KS}} = 0.00406$ and $p_t = 0.0327$) and OM ($p_{\mathrm{KS}} = 0.0276$ and $p_t = 0.0402$), but not so for XM ($p_{\mathrm{KS}} = 0.0861$ and $p_t = 0.282$).

differences (at the $\alpha = 0.05$ level) in the distribution of values in terms of a two-sided Kolmogorov–Smirnov test ($p_{\mathrm{KS}}$) and in terms of a $T$-test ($p_t$) for both BM and OM, whereas the differences in XM are not considered to be significant.