[Reviews · NeurIPS 2020]

Review 1

Summary and Contributions: Carlsson’s TDA is one of the most elegant analysis for deciding on the most stable topology of a cloud of points. Here, that theory is used in an innovative manner to study task-evoked within-subject brain state similarities (here expressed as clusters) over the decoded topology. This construct is associated by the authors to the construct of functional connectivity (ln 34) although some argumentation can be made regarding whether the two constructs are analogous or the same. Regardless of the philosophical discussion, this is an excellent paper; with rock solid maths, replicable experiments, strong and clear results, and conclusions commensurate with evidence provided. It is my understanding, that the distinctive key to the proposal is (i) the chosen projection to the cubical complex as this defines the topology from where the filtration of the simplicial complex is derived, and (ii) the trajectory analysis, which encodes the solution to the neuroscientific question at hand. Although, the method is exemplified to capture age differences, it is easy to see how the proposed method can be reused to explore other questions. In terms of innovation, this paper is arguably the best of my reviewing lot for this year. Moreover, the paper makes a critical statement: the simplicial complex generalizes the graph (ln 64). This is known -no news here-, yet the remark is extremely important in my opinion; despite all the excellent contributions to understanding brain function based on graph theory, the graph (any kind!) is insufficient to express all the complexity of the brain integrational activity and thus more general mathematical objects must be explored.

Strengths: + The maths in section 2 makes an excellent effort to convey the essence of TDA, which is otherwise an extremely sophisticated theory. The complement in A.1 is superb. + This is a very innovative work.

Weaknesses: + The link between the construct explored and “classical” neuroscientific questions e.g. connectivity or other is weak. This could be strength, if indeed it is a new construct with value for neuroscience, but this is not shown. + Synthetic verification and validation is lacking.

Correctness: The underpinning TDA theory is very strong mathematically. Theorem 1 (stability) is relatively simple, and the proof in the supplementary material is virtually trivial. So unless I’m missing something, as far as I can tell, the theory is correct. Experimentally, the authors have opted for a relatively simple design which also add to the strength of the paper and the message. The dataset is external and hence, control over its acquisition is limited, but this is in principle irrelevant for the hypothesis made.

Clarity: Given the complexity of the underpinning theory (TDA), the clarity is excellent in my opinion (although I may be biased in the sense that I am fairly familiar with TDA). A novice may struggle to follow. The rest of the paper, introductions, experiment description and results are also easy to follow again if you are somewhat knowledgeable in the field. Perhaps only some more detail on the description of the processing of the MRI dataset is missed e.g. specific functions and parameterizations. (Note that this is not the description of the fMRI dataset, which can be found in Supp. Material A.3) Figures labelling in some cases are unreadable because of font size (sorry, I’m old and my sight is starting to fail!). Since some of the colors are only for aesthetics purposes e.g. Fig 2c, FigA.5, etc they can be perhaps be avoided to avoid potential confusion.

Relation to Prior Work: The relation to previous work is perhaps one of the weakest parts of the paper. However, the text is very dense throughout the full draft -the authors succeed in saying a lot with very few words-, hence, I believe that they might have sacrificed a bit of the referential framework, to have more room for presenting their research which is a totally understandable decision.

Reproducibility: Yes

Additional Feedback: + Something that I appreciate from graph theory analysis in neuroscience, is that most properties of the graph have a clear neuroscientific interpretation. Bringing a new object to the field, requires some time to establish such a “dictionary” between mathematical properties and associated domain interpretations. Here, the curvature analysis is fascinating from a mathematical point of view, and it is brilliantly used to discriminate groups. However, prediction does not necessarily go hand in hand with explanation, and I reckon there would be other mathematical properties of the trajectory object that would be expressive of such group differences. Is there any translational meaning of the curvature to neuroscience? + Only face validity is established. Other types of validity could have been attempted -although I reckon the authors are keeping this for a journal-. + The relevance of the work in my opinion is beyond doubt. Notwithstanding, it is difficult not to wonder whether a simpler approach would have perhaps succeeded in answering the question at hand. No effort is made to show whether this is the case or not. For instance, in the coarsest sense, classification of age groups from the observations could have been addressed from a classical classification approach leading also to significant differences. =========== After Rebuttal =========== After seeing the rebuttal by the authors, the discussion on this paper has been open. If I am interpreting my fellow reviewers' opinion correctly, it is my feeling that while we did not reach to a full agreement -rejection position was not too strong, but neither push for oral-, but at least paper acceptance can be recommended.


Review 2

Summary and Contributions: The authors proposed to apply time-varying persistence diagrams from algebraic topology on fMRI volumes to show that these topological representations are capable of capturing age-related differences between adults and children.

Strengths: - This work makes use of the topological representations from persistence diagrams which are robust to noise and variability among individuals, which is an interesting approach. - Being the first to apply topological data analysis directly on fMRI data, as claimed by the authors.

Weaknesses: - Persistent homology on brain topology has been studied before, e.g., Chung et al, Persistent homological approach to detecting white matter abnormality in maltreated children: MRI and DTI multimodal study, and many other papers from his group. What is the difference between Chung's work and this one and what novelty does this paper provide other than going from network to 4D? - What is the dimension of the real data? - The paper lacks detailed explanation in the neuroscience background, which supposed to be a very important piece in this paper.

Correctness: Yes

Clarity: Not quite. It is a little difficult to follow

Relation to Prior Work: Not quite.

Reproducibility: Yes

Additional Feedback: Authors mainly apply the existing methods from algebraic topology and seems to lack critical contributions. Even as an application paper, it is not very clear why this work is significantly important in the neuroscience domain.


Review 3

Summary and Contributions: The paper applies persistent homology with cubical complex to time-varying fMRI data. The idea is to use the whole brain fMRI image (with whole brain mask or ROI masks applied) as the filter function for the computation of persistent homology. And then the persistence diagrams over different time points are used to differentiate age groups. A first study uses the total persistence measurement (one scalar value per diagram) over different time points. A second study use persistence image which maps the diagrams into vectors. Both cases show that the topological feature are able to differentiate the age groups.

Strengths: The result seems scientifically significant. The usage of topological methods is well-thought and properly carried out. The experiments are well executed. Baseline methods on non-topology descriptors for fMRI are compared with. The research clearly will lead to interesting discovery of the data.

Weaknesses: My main concern is that the novelty of the methodology is very limited given abundant previous applications of persistent homology to various images (including fMRI). There is a long list of previous results on applying persistent homology to fMRI, structural MRI (mostly resting-state though), and EEG data (the first published in 2009, "Persistence Diagrams of Cortical Surface Data", IPMI 2009). These methods should have been cited and compared with. I do agree that the findings over the dataset can be potentially impactful. And I think the paper is quite well-written. However, I think its value is only in the application of the method to this particular dataset and the novel domain-specific insights. I do not think this paper fits the NeurIPS conference quite well. it seems to be a better fit to a neuroscience venue such as NeuroImage, Human Brain Mapping, etc. After reading the rebuttal and other reviews, I am raising my score but I am staying on the negative side. This paper is a good one for the application of persistent homology to this particular long-term task fMRIs. I do not think it is a good fit to NeurIPS if we were to judge by the methodology part. But I am partially convinced by my fellow reviewers that if NeurIPS were to have any paper on the neuroscience track, this should be one. The idea of using cubical persistence in the imaging context is straightforward, although I have not seen any methods using cubical complex on fMRIs. One thing I would encourage the authors to add to the final version of the paper: the baselines (baseline-PP and PP-TDA) are only taking average values within 100 ROIs (an 100-times dimension reduction), also they only use the correlation between ROIs. Meanwhile, persistent homology is using the full voxel image. To be more convincing, the authors should use the actual full fMRI image (like MVPA). If the data samples are insufficient, the authors should use average/max/min values of each ROIs as features and show that persistent homology features outperform these voxel-value-based features.

Correctness: Yes.

Clarity: Yes. The paper is clearly written.

Relation to Prior Work: No. While the authors tried to cover quite some papers from original persistent homology and from recent progress in applying persistent homology to learning. It seems that they missed a whole literature of persistent homology applied to brain imaging.

Reproducibility: Yes

Additional Feedback: The implementation details are well presented and the code is (or will be) shared. I really appreciate it. I am a bit curious about the time-series visualization tool. Perhaps some brief explanation in the paper can be helpful.

[Author Response · NeurIPS 2020]

We thank the reviewers for their positive, kind, and constructive feedback! The two main points of criticism concerned (1) the lack of neuroscience background information, and (2) missing discussion and comparison to prior work. Due to page limitations, these points were insufficiently addressed. In a revision, we will provide additional details here as well as more citations concerning prior work in TDA and neuroscience.

**Differences/comparison to previous work**: We are the first TDA paper to work *directly* with fMRI input data (using cubical complexes). Prior work uses auxiliary representations such as networks extracted from correlation matrices [2, 7]. Moreover, previous studies often use other measuring modalities such as structural MRI for anatomical analyses [1], or diffusion MRI/DTI for studying white matter integrity [2]. Our cubical persistence formulation is the first of its kind in the context of functional MRI measuring brain activity during a movie-watching task. We will make this delineation more clear in a revision.

Inspired by this feedback, we also prepared experiments using a more conventional formulation of TDA methods based on correlation graphs of the data, which we created from correlation matrices using a correlation distance filtration [3] (terminology follows the paper; time-based: TT-TDA; voxel-based, parcelled: PP-TDA). The new values in the table are highlighted; the remaining rows are duplicated from the table in the paper. We observe that the time-based correlation matrix/graph is improved by topological feature extraction, while the voxel-based (parcelled) correlations are not improved. This demonstrates the advantage of using the input data directly, instead of requiring auxiliary representations.

| Method | BM | OM | XM |
|---|---|---|---|
| BASELINE-TT | 0.09 | 0.02 | 0.24 |
| BASELINE-PP | 0.41 | 0.40 | 0.40 |
| SRM | 0.44 | — | — |
| TT-TDA | 0.16 | 0.08 | 0.24 |
| PP-TDA | 0.19 | 0.24 | 0.23 |
| $\|\mathcal{D}\|_1$ | 0.46 | 0.67 | 0.48 |
| $\|\mathcal{D}\|_\infty$ | **0.61** | **0.77** | **0.73** |

Performance of baselines, standard TDA approaches, and our method for the age prediction task.

**Reviewer 1**: Thank you very much for your exuberant feedback, we really appreciate it! Concerning the weaknesses that you mentioned: indeed, there is no direct 'link' between our approach and classical approaches; however, our topological features 'live' in the original space of the data and can be localised [5, 8], i.e. endowed with a minimal geometry. Other topological approaches, which use correlation graphs as intermediaries, do not have features that directly relate back to the data. We plan to explore this in the future and we are convinced that our approach will also open up other avenues of inquiry. We will also run our approach on synthetically-generated data sets [4] for verification and validation (in order to study the limitations of our approach).

**Reviewer 3**: Thank you very much for your positive feedback! Concerning the weakness that you mentioned, please see our general points above—in a revision, we will delineate this work better from related papers. Thanks for the links to additional papers; we will cite and discuss them accordingly. • *Dimensions of the data*: Thanks for highlighting this; we will add it to the paper! The 4D volume of each participant has dimensions $65 \times 77 \times 60 \times 168$ (as described in the paper, we are not considering all 168 time steps). • *Neuroscience background*: We will add an appropriate section to the paper or the supplemental materials. • *Contributions*: We are the first work utilising cubical persistent homology in the context of fMRI. We use theoretical tools with a strong mathematical foundation and apply them in a novel way. This results in *dynamic* representations (previous work only considered static representations), the brain state trajectories, whose calculation combines topological features with the diffusion geometry method PHATE [6], yielding a novel set of features that were previously not considered in the literature. As we show in the paper, these trajectories are capable of capturing the dynamics of cognition. We will emphasise these contributions more in the revision.

**Reviewer 4** Thank you very much for your constructive feedback! We will delineate our work (novel cubical persistence calculations) better from existing work (requiring auxiliary representations such as networks). • *Generalisability*: While we focussed on one data set for this initial submission, our method can be applied to *any* neuroimaging data set. A key feature is its abstraction—as we show in the prediction task, this may help counteract noise and intra-subject variability. We are convinced that other applications in neuroscience, using metrics other than age prediction, can benefit from our approach (which is why we will make all code available). Thus, we think that our work paves the way for a different sort of topology-based neuroscience methods that are based on *direct* feature extractions from the data. We will explore the feasibility of our work on other modalities (such as EEG) in the future (presently, they are out of scope for this work). • *Time series visualisation*: We will extend the description of this approach and provide examples.

[1] M. K. Chung et al. Persistence diagrams of cortical surface data. In *Information Processing in Medical Imaging*, pages 386–397. Springer, 2009.

[2] M. K. Chung et al. Persistent homological sparse network approach to detecting white matter abnormality in maltreated children: MRI and DTI multimodal study. In *Medical Image Computing and Computer-Assisted Intervention – MICCAI 2013*, pages 300–307. Springer, 2013.

[3] M. K. Chung et al. Topological distances between brain networks. In *Connectomics in NeuroImaging*, pages 161–170, 2017.

[4] C. T. Ellis et al. Facilitating open-science with realistic fMRI simulation: validation and application. *PeerJ*, 8:e8564, 2020.

[5] J. Erickson and K. Whittlesey. Greedy optimal homotopy and homology generators. In *Proc. SODA*, pages 1038–1046. SIAM, 2005.

[6] K. R. Moon et al. Visualizing structure and transitions in high-dimensional biological data. *Nature Biotechnology*, 37(12):1482–1492, 2019.

[7] M. Saggar et al. Towards a new approach to reveal dynamical organization of the brain using topological data analysis. *Nature Communications*, 9(1):1399, 2018.

[8] A. Zomorodian and G. Carlsson. Localized homology. *Computational Geometry*, 41(3):126–148, 2008.


[Meta-Review · NeurIPS 2020]

All 3 expert reviewers agree that this paper is interesting, relevant and very well carried out. There was some discussion as to whether or not NeurIPS was the right venue for the paper due to its applied nature, but at the end both the reviewers and AC agree that this is an excellent paper for the neuroscience/neuroimaging track.